# Acoustically manipulating internal structure of disk-in-sphere endoskeletal droplets

Gazendra Shakya [1,4], Tao Yang [1,4], Yu Gao[1], Apresio K. Fajrial [1], Baowen Li [1], Massimo Ruzzene [1], Mark A. Borden[1,2,3] & Xiaoyun Ding [1,2,3✉]

Manipulation of micro/nano particles has been well studied and demonstrated by optical, electromagnetic, and acoustic approaches, or their combinations. Manipulation of internal structure of droplet/particle is rarely explored and remains challenging due to its complicated nature. Here we demonstrated the manipulation of internal structure of disk-in-sphere endoskeletal droplets using acoustic wave. We developed a model to investigate the physical mechanisms behind this interesting phenomenon. Theoretical analysis of the acoustic interactions indicated that these assembly dynamics arise from a balance of the primary and secondary radiation forces. Additionally, the disk orientation was found to change with acoustic driving frequency, which allowed on-demand, reversible adjustment of the disk orientations with respect to the substrate. This dynamic behavior leads to unique reversible arrangements of the endoskeletal droplets and their internal architecture, which may provide an avenue for directed assembly of novel hierarchical colloidal architectures and intracellular organelles or intra-organoid structures.

[1] Paul M. Rady Department of Mechanical Engineering, University of Colorado, Boulder, CO 80309, USA. [2] Biomedical Engineering Program, University of Colorado, Boulder, CO 80309, USA. [3] Materials Science and Engineering Program, University of Colorado, Boulder, CO 80309, USA. [4] These authors contributed equally: Gazendra Shakya, Tao Yang. ✉email: Xiaoyun.Ding@Colorado.edu

Manipulating particles from nanometer scale to micrometer and millimeter scale, such as molecules, cells, colloids, droplets, and small model organisms, has been important in biology, chemistry, and medicine. Micromanipulation can be achieved by optical, electromagnetic, mechanical, acoustic approaches or their combinations. For instance, optical and nanorobotic methods have been successfully used for manipulation in nano and micro scale[1–3]. Electrical and acoustic methods have achieved high throughput particle manipulation for applications such as bioparticle patterning, protein characterization, cell separation and droplet sorting, etc[4–6]. However, an advanced manipulation of the internal structure of complex colloids has been a long-standing challenge and rarely reported so far.

Endoskeletal droplets are a class of complex colloids containing a solid internal phase cast within a liquid emulsion droplet. In an early example, petrolatum-in-hexadecane droplets were shown to have controllable shape by microfluidic processing and manipulation of the balance between internal elasticity and surface tension[7]. Such droplets have been used to form fluid networks[8], which change their orientation and shape in response to external stimuli[9,10]. Recently, solid hydrocarbon in liquid fluorocarbon was described, in which melting of the internal solid hydrocarbon phase triggered vaporization of the fluorocarbon liquid phase[11]. Additionally, solid-in-liquid perfluorocarbon droplets were made that exhibited a unique droplet structure: a fluorocarbon solid disk suspended inside a liquid fluorocarbon droplet[11].

Here, we describe the synthesis of monodisperse disk-in-sphere perfluorocarbon droplets and their internal structure manipulation in an acoustic field. Among the host of attractive and repulsive interactions available to colloidal particles[12–15], acoustic radiation forces, induced by acoustic fields, have proved to be an efficient way to manipulate particles. In an acoustic field, the motion of the particles (much smaller than the wavelength of the acoustic wave) suspended in an aqueous medium is driven by two main acoustofluidic forces: acoustic radiation force, which is responsible for the motion of larger particles ($>2\,\mu m$), and the stokes drag force arising from acoustic streaming, which is responsible for the motion of much smaller particles ($<2\,\mu m$)[16–18]. Even though the particle responses to acoustic radiation forces have been used for applications, such as particle separation[19–22], particle manipulation[23,24] and assembly of complex structures[25,26], the manipulation of internal structures within endoskeletal colloids has not yet been reported. In this work, two unique phenomena were discovered. First, we observed that the droplets clustered in such a way that the disks oriented orthogonally to the cluster centroid. Second, we found that the disk orientation could be manipulated by changing the frequency of the acoustic waves. These behaviors can be described by an investigation of their acoustic interactions and a balance of the primary and secondary radiation forces, and they offer the tantalizing possibility of acoustically constructed colloidal assemblies, organelle manipulation, and organoid structures with dynamic and tunable internal structures.

## Results and discussion

**Microfluidic fabrication of endoskeletal droplets**. Disk-in-sphere endoskeletal droplets consisting of solid perfluorododecane (PFDD, $C_{12}F_{26}$, density $\rho_s = 1.73\,g/cm^3$ and sound velocity $c_s = 641\,m/s$[22]) and liquid perfluorohexane (PFH, $C_6F_{14}$, $\rho_l = 1.69\,g/cm^3$, $c_l = 479\,m/s$[27]) were fabricated using a flow-focusing microfluidic channel (Fig. 1a, b, c). Detailed information about the device fabrication and droplet fabrication can be found in the methods section. The device was selectively heated to melt the solid component (45 mol%) prior to droplet generation. Thus, initially isotropic

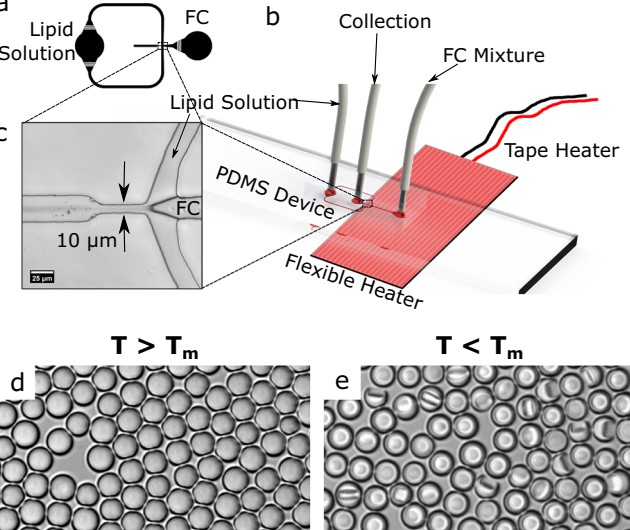

**Fig. 1 Microfluidic fabrication of endoskeletal droplets. a** Design of the Microfluidic Channel used to generate droplets. **b** Droplet generation schematic showing the PDMS device and different inlets and outlets. Zoomed in image of the flow-focusing junction is shown in **c** Scale bar, 25 μm. **d** Endoskeletal droplets generated using this technique at a higher temperature ($T > T_m$) are single-phase liquid droplets where the solid disks are all melted. When the droplets are cooled to a lower temperature ($T < T_m$), the solid phase separates and forms the endoskeleton confined by the droplet boundaries (shown in **e**). The disks are randomly oriented with different orientations. Parallel orientation and perpendicular orientation of the disk are shown in the image inserts where the disks are seen as circle and rectangle respectively. Scale bar, 20 μm.

liquid droplets of uniform size (radius $a = 4.78 \pm 0.25\,\mu m$) (Supplementary Fig. 1a) were produced that contain a mixture of the solid and liquid fluorocarbons (FC) (Fig. 1d) in a liquid state at ~42 °C. When cooled, the PFDD solidified to create the unique disk-shaped structure (radius $R = 3.35\,\mu m$, thickness $h = 2.4\,\mu m$) inside each individual droplet (Fig. 1e). Although the disks were confined inside the liquid PFH droplets, they were observed to rotate and translate within the confinements of the droplet (Supplementary Fig. 1b). The orientation is denoted as 'parallel' and 'perpendicular' with respect to the substrate. In the parallel orientation, the disks appeared as circles, while in the perpendicular orientation the disks appeared as rectangles, as shown in the inserts in Fig. 1e.

**Acoustically driven endoskeletal droplet patterning**. We incorporated two counter-propagating surface acoustic waves (SAWs) with center frequency of 20 MHz, generated by applying voltage to interdigital transducers (IDTs) deposited on a piezoelectric material (Lithium niobate, LiNbO3) (Fig. 2a). These SAWs traveled through an aqueous phase, where the droplets were suspended, confined in a polydimethylsiloxane (PDMS) channel (2 mm width, 18 μm height) attached to a Lithium Niobate substrate. The counter-propagating acoustic waves formed one-dimensional (1D) standing surface acoustic wave (standing SAW) in the x-direction within the channel (Fig. 2a).

We observed the endoskeletal droplets suspended in this standing SAW field. The droplets attracted one another and moved to form clusters of various sizes (Supplementary Movie 1). Figure 2b–g shows an example of these clusters arranged under

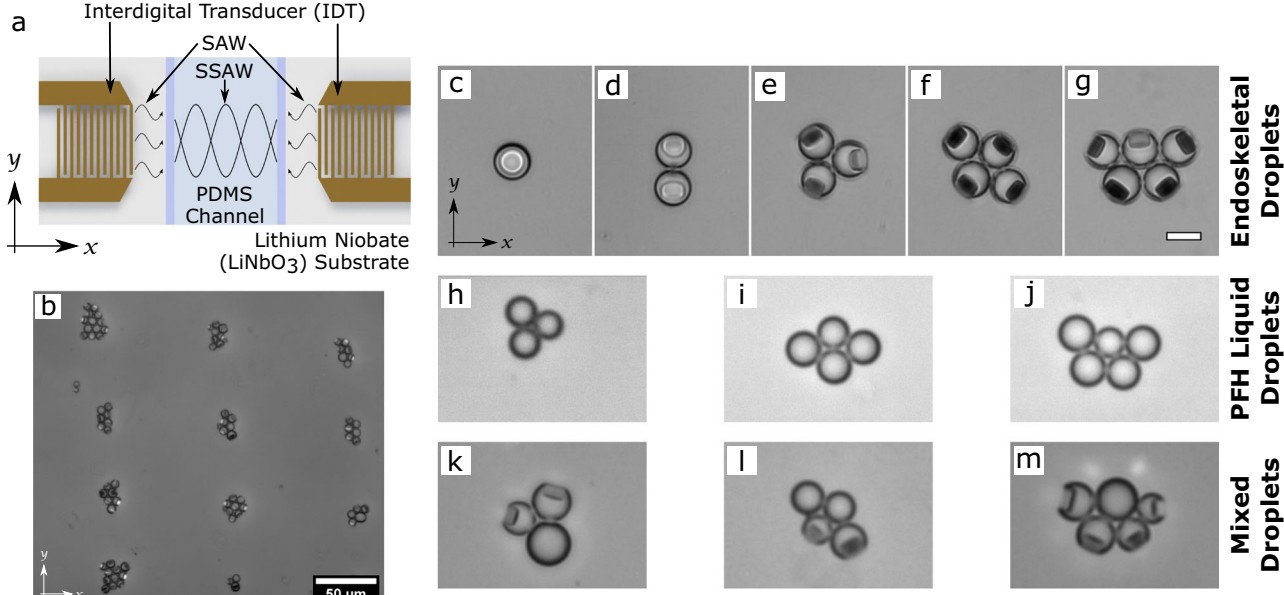

**Fig. 2 Endoskeletal Droplet Patterning under standing SAW. a** Schematic of the SAW IDT devices used for patterning experiments. **b** Clustering behavior of endoskeletal droplets under 1D standing SAW. Scale bar, 50 μm. Zoomed in images of the droplet clusters under 1D standing SAW with 1 droplet (**c**), 2 droplets (**d**), 3 droplets (**e**), 4 droplets (**f**), 5 droplets (**g**). Note that the disk orientations are different for smaller clusters (**c** and **d**) compared to larger clusters (**e**–**g**). **h**, **i**, **j** Liquid PFH droplet show similar clustering behavior as endoskeletal droplets. **k**, **l**, **m** Clusters containing mixture of liquid only PFH droplets and endoskeletal droplets. Note that the clustering phenomenon is the same for all three types of droplets clusters and the disk orientations are not affected by the absence of other disks as well. Scale bar for all individual cluster images, 10 μm.

the standing SAW. For single droplets (monomers), the disk was found to be parallel to the substrate and pushed up to the top of the droplet (Fig. 2c, Supplementary Movie 2). For 2-droplet clusters (dimers), the orientation of the disk was midway (~45°) between the parallel and the perpendicular orientations (Fig. 2d, Supplementary Movie 3). For clusters containing more than 2 droplets, the disks were oriented perpendicular to the substrate (Fig. 2e–g, Supplementary Fig. 2). Interestingly, for these larger clusters, the disks always arranged such that the normal of the basal plane of each disk pointed to the centroid of the cluster (Supplementary Fig. 2).

Furthermore, the solid disk inside a droplet only rearranged when it came close to another particle. This can be seen in Supplementary Movie 1, where the disks begin rearranging as the droplets come into close proximity of each other. This neighbor-dependent behavior suggested an interaction force between the approaching particles. To better understand this behavior, we turned to droplets without internal structures.

**Acoustic assembly of liquid only fluorocarbon (FC) droplets.** Liquid-only PFH droplets of similar size (~5 μm radius) as the endoskeletal droplets were generated using the same microfluidic setup and suspended in the same SAW setup (Fig. 2a). The liquid-only PFH droplets were observed to cluster in a similar manner as the endoskeletal PFDD-in-PFH droplets (Fig. 2h–j). Samples containing a mixture of PFH and endoskeletal droplets were also observed to cluster in a similar manner (Fig. 2k–m). This behavior indicated that the disks themselves did not cause the droplets to cluster. Moreover, the disks arranged in similar orientations, regardless of whether the neighboring droplet contained a disk (Fig. 2e–m). This result indicated that the disk rotation and orientation was not caused by neighboring disks.

**Theoretical investigation of clustering behavior.** Acoustic radiation forces are responsible for particle motion in an acoustic field. The primary radiation potential experienced by the droplets

due to acoustic pressure gradients in the limit of $a \ll \lambda_w$ (where $\lambda_w$ is the acoustic wavelength in water and is ~74 μm for 20 MHz), also called Gor'kov potential[28], is given by[29,30]:

$$U_{rad} = V_p \left( \frac{1}{2} f_{0,l/w} \beta_w \langle p_1^2 \rangle - \frac{3}{4} f_{1,l/w} \rho_w \langle v_1^2 \rangle \right) \quad (1)$$

Here, $f_{0,l/w} = 1 - \frac{\beta_l}{\beta_w}$ and $f_{1,l/w} = \frac{2(\rho_l - \rho_w)}{2\rho_l + \rho_w}$ (2a, b)

where $V_p$ is the droplet volume, $f_{0,l/w}$ and $f_{1,l/w}$ are the monopole and dipole scattering factors ($l/w$ for PFH liquid/water system), $\beta = \frac{1}{\rho c^2}$ is the compressibility, $\rho$ is the density, $c$ is the speed of sound, subscript $l$ signifies PFH liquid and subscript $w$ signifies water, $p_1$ is the first order pressure amplitude, $v_1$ is the particle velocity and $\langle \cdots \rangle$ stands for the time average. For a 1D standing acoustic wave, i.e., incoming acoustic pressure $p_{in} = p_0 \cos(k_w x)$ (where $p_0$ is the pressure amplitude and $k_w = \frac{2\pi}{\lambda_w}$ is the wavenumber in water), the primary radiation force $F_{pri}$ can be simplified as[29,30]:

$$\boldsymbol{F_{pri}} = -\nabla U_{rad} = \frac{1}{2} V_p E_0 \Phi_{l/w} k_w \sin(2k_w x) \hat{\mathbf{x}} \quad (3)$$

where $E_0 = \frac{1}{2} \beta_w p_0^2$ is the acoustic energy density and $\Phi_{l/w} = f_{0,l/w} + \frac{3}{2} f_{1,l/w}$ is the acoustic contrast factor[17,29–31]. The sign of the acoustic contrast factor determines the direction of the primary radiation force (towards pressure node or antinode) that a particle experiences at its location within the standing SAW field. For our droplet emulsion (PFH liquid in water), $\Phi$ is negative (−4.26), hence the PFH droplets were driven to the pressure antinodes (Fig. 3a).

When one droplet approaches another as they migrate to the antinode, the acoustic wave scattering from the neighboring body induces an additional interaction force, called the secondary radiation force[31]. The interaction energy at the anti-nodal line between two particles with negative contrast factor (two PFH

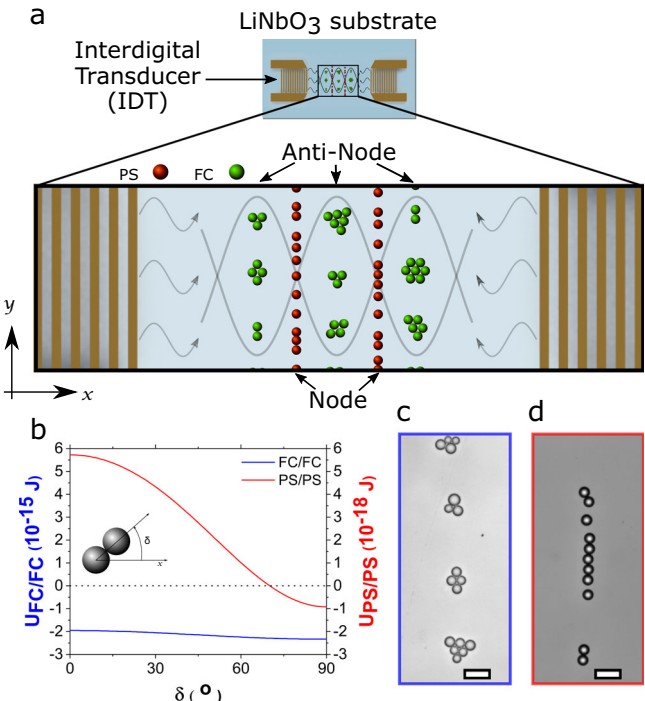

**Fig. 3 Liquid PFH droplet clustering and radiation forces. a** Schematic of the SAW device shown with the assembly type and location of liquid PFH droplets and the PS beads of similar sizes. **b** Secondary interaction energy between two contacting PFH droplets (red line, Eq. 4) and PS beads (blue line, Eq. S3) with orientation angle δ with respect to wave propagation direction, i.e., x-axis. Positive values indicate repulsion and negative values indicate attraction. Source data are provided as the source data file. **c** Assembly of liquid PFH droplets under 1D standing SAW compared to the assembly of PS particles under 1D standing SAW (shown in **d**). Note that PS particles form chains whereas PFH particles forms clusters as shown by **b**. Scale bar, 20 μm.

droplets in this case) in the anti-nodal plane under a 1D standing wave was simplified[31] as:

$$U_{sec} = \frac{V_p^2 E_0 k_w^2}{8\pi r}(3f_{0,l/w}f_{1,l/w}\cos^2\delta - 2f_{0,l/w}^2) \quad (4)$$

where $r$ is the distance between two droplet centers and $\delta$ is the orientation angle between two droplets (Fig. 3b) with respect to the wave propagation direction (x-axis). The secondary radiation force ($F_{sec} = -\nabla U_{sec}$) derived from Eq. 4 is consistent with the well-known Bjerknes forces for gas bubbles in short-range[32]. The calculated secondary interaction energy from Eq. 4 between two contacting PFH droplets (i.e., $r = 2a$) is shown in Fig. 3b (blue). Since the magnitude of $f_{0,l/w}$ (4.74) is about one order larger than $f_{1,l/w}$ (0.32), the monopole effect dominates and the interactions between two spheres are almost isotropic attractions (Supplementary Fig. 3a). As a result, two-dimensional closed packed clusters are formed (Figs. 2h–j, 3c). This type of two-dimensional (2D) clustering behavior is usually observed in polystyrene (PS) spheres[33], cells[33,34] and silicone microspheres[14] under two-dimensional (2D) standing waves (instead of a 1D standing wave) where acoustic waves propagate in both the $x$ and $y$ directions. Particles with a positive acoustic contrast factor behave differently in a one-dimensional (1D) standing wave. For comparison, the interaction energy of two contacting polystyrene (PS) particles ($f_{0,s/w} = 0.46$, $f_{1,s/w} = 0.038$, and $\Phi_{s/w} = 0.517$) with the same size placed at the nodal plane (Eqn. S3) of a one-dimensional (1D) standing wave was also plotted in Fig. 3b (red).

For PS particles, dipolar interactions with preferential angle parallel to the nodal lines are expected (Supplementary Fig. 3b), which explains why colloidal chains of PS particles were formed on our standing SAW device (Fig. 3d), and as seen in similar studies done on PS particles by Shi et al.[33] and Vakarelski et al.[35].

**Interpreting disk arrangement behavior.** From the previous discussions, we can conclude that the aggregation of endoskeletal droplets, as well as liquid PFH droplets were due to the combined effects of primary and secondary radiation forces. To better understand the unique orientations of the solid disks inside the endoskeletal droplets, we performed finite element simulations using COMSOL (version 5.0) where the forces and torques on the inner disk were calculated the following prior work[29,36,37] (see Supplementary Section 2, Supplementary Fig. 4). The equilibrium disk orientation was determined by the zero-torque configuration and is shown in Fig. 4a, b. Here, the angle $\theta_i$ is defined as the angle between the disk (of $i$th droplet) and the nodal line (positive y-axis), whereas the angle $\alpha$ is defined as the angle between the nodal line and the line joining the centroid and the center of the 1st droplet (which determines the orientation of the whole cluster). The equilibrium angles of the disks are given by $\theta_i$ values at zero torque. These calculated equilibrium angles (Fig. 4b) are consistent with the orientations of disks seen in experiments (Fig. 2e). These equilibrium angles were also calculated at different cluster orientation angles ($\alpha$), which is shown in Supplementary Fig. 5 along with the experimentally observed $\theta$ values. Similar equilibrium angles at zero torque configurations were calculated for different droplet clusters (Supplementary Table 1, Supplementary Fig. 6), which match the experimental observations as well.

Note that the results shown in Fig. 4a, b assume the disks to be at the droplet centers and not translating (only rotating). In reality, as the inner PFDD solid has a positive acoustic contrast factor against surrounding PFH liquid ($f_{0,s/l} = 0.45$, $f_{1,s/l} = 0.016$, and $\Phi_{AC,s/w} = 0.48$), disks are expected to be pushed to the edges of the droplets where local pressure minima exist (Fig. 2d–g, Supplementary Fig. 2, Supplementary Movie 1). However, translation did not affect the equilibrium disk orientation angles. This was shown by performing disk dynamics simulations that show behavior similar to what was seen in experiments, where the disks were pushed to the far edges of the droplets but retained their equilibrium angles (Supplementary Movie 4 and see Supplementary Section 3).

Once the equilibrium disk position and orientation were determined from the dynamic simulation, the local Gor'kov potential ($U_{rad}/V_p$) density and radiation force density ($F_{pri}/V_p$) on the disks inside droplets of a cluster were calculated and is shown in Fig. 4c–i. The local forces pointing from the droplet center to the edge explains the origin of disk movement and final positions for multi-droplet clusters.

However, the differences in disk alignment seen in monomers and dimers suggest additional wave contributions. Due to the wave velocity differences in water and the solid substrate, the surface wave generated from the IDT's traveled into the water (i.e., "leaky" waves) at the Rayleigh angle $\theta_R = \text{acos}\left(\frac{k_{LN}}{k_w}\right)$, where $k_{LN} = \frac{2\pi}{\lambda_{LN}}$ is the wavenumber in the solid lithium niobate substrate (Fig. 4j). Thus, these two counter-propagating leaky waves formed a quasi-standing wave in the z-axis along with forming a standing wave in the x-axis (Fig. 4j) (Supplementary Section 4). As a result, a more realistic pressure distribution can be approximated as: $p_{in}' = p_0 \cos(k_{LN}x)\cos(k_w \sin_{\theta_R} z)$[38]. This pressure amplitude distribution was also confirmed with numerical simulation results (Supplementary Fig. 8), taking the leaky wave component into consideration following Nama

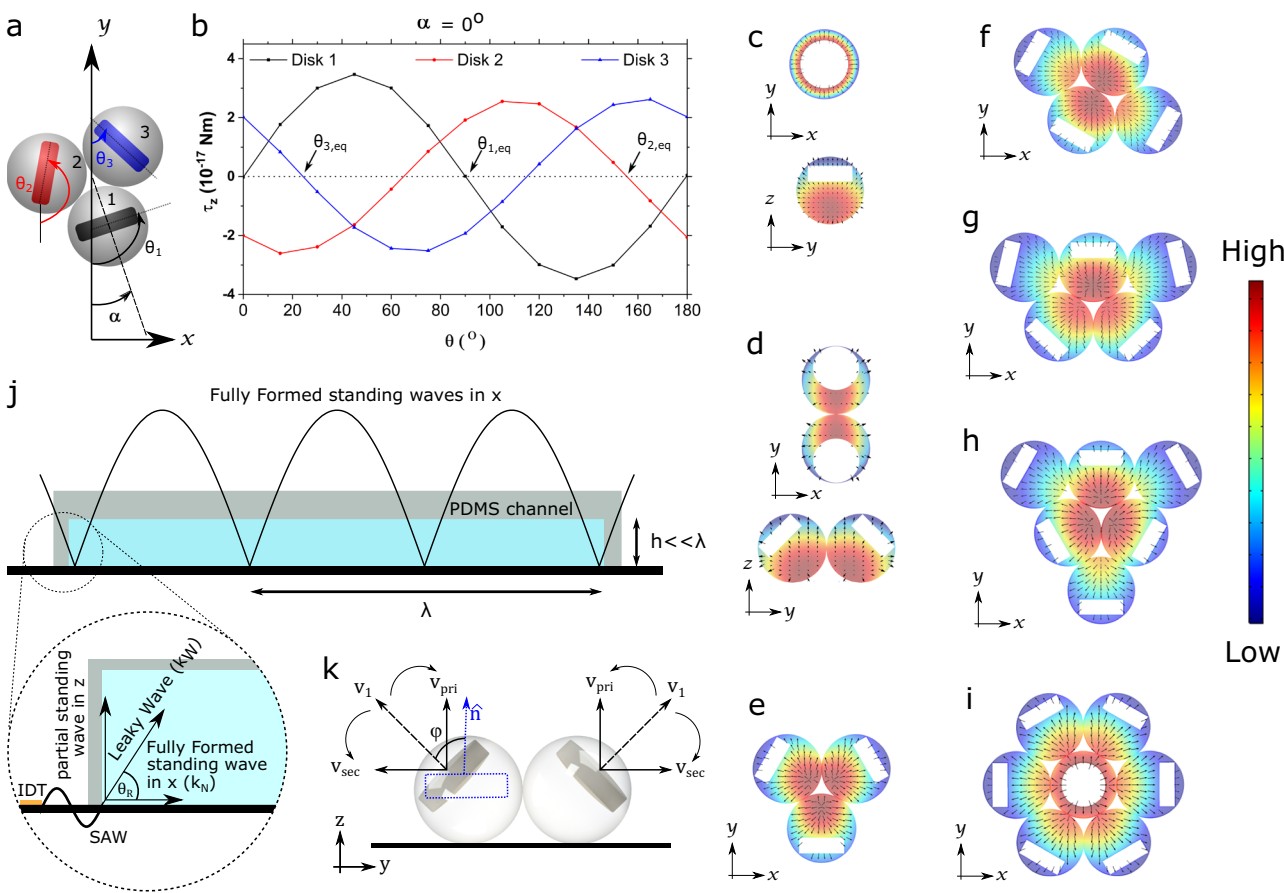

**Fig. 4 Understanding disk orientation behavior. a** Schematic of disk orientation angles inside a trimer system plotted in **b. b** Torque $\tau_z$ on inner disks inside a trimer cluster vs. disk orientation angle from COMSOL simulations. The equilibrium angles $\theta_{eq}$ (shown by arrows), i.e., equilibrium orientations are determined by the zero torques with negative slope. Source data are provided as a source data file. **c.** Simulation results showing the effect of standing SAW on clusters with 1 droplet (**c**), 2 droplets (**d**), 3 droplets (**e**), 4 droplets (**f**), 5 droplets (**g**), 6 droplets (**h**) and 7 droplets (**i**). The simulation results match the experimental observations of the droplet clusters shown in Fig. 2c–g, Supplementary Fig. 2. **j** Schematic showing the generation of fully formed standing wave in the x-direction and partial standing wave in the z direction (due to restricted channel height). **k** Schematic showing the disk orientation inside the droplet is a result of the interplay between the primary and secondary radiation forces. φ is the angle between $v_1$ and the disk axis (shown in dotted blue for one of the droplets, with a random orientation, in a non-equilibrium position). At equilibrium torque, the disk radial symmetric axes should align with particle velocity $v_1$ direction (φ = 0), which is a sum of contributions from primary radiation $v_{pri}$ (along z-axis for all disks) and the secondary radiations $v_{sec}$ (along radial direction away from the droplet cluster, i.e., y-axis for a two-droplet cluster).

et al.[39], Devandran et al.[40] and Barnkob et al.[41]. The additional standing wave component in the z-axis generated additional torque, which aligned the inner disks parallel to the *xy* plane, as in a single droplet. When simplified as a Raleigh disk ($R \ll \lambda_w$)[42], the torque on the disk in x-axis was shown as[43]:

$$\tau_x = -\tfrac{4}{3}\rho_l R^3 v_{rms}{}^2 \sin(2\varphi) \qquad (5)$$

where $v_{rms}$ is the root-mean-square of particle velocity $v_1$ and the angle $\varphi$ is the angle between $v_1$ and the orientation of the disk (Fig. 4k). As a result, the equilibrium disk orientation was along the particle velocity direction, i.e., φ = 0. The particle velocity $v_1$ is the sum of contributions from the primary radiation $v_{pri}$ and secondary radiation $v_{sec}$ vectors, i.e., $\boldsymbol{v_1} = \boldsymbol{v_{pri}} + \boldsymbol{v_{sec}}$. The particle velocity from primary radiation on each disk was the same, as shown by $\boldsymbol{v_{pri}} \sim \frac{p_0}{\rho_l c_l} k_w \sin_{\theta_R} a\hat{\mathbf{z}}$ (see Supplementary Section 4). For a monomer where there is no secondary radiation force contribution from a neighboring droplet, the effect of the leaky wave was observed in the disks, which rose to the top and oriented parallel to the substrate. This was shown both in

experiments (Fig. 2c, Supplementary Movie 2), as well as dynamic simulations (Fig. 4c, Supplementary Movie 5). In case of the dimer, the disk orientation was balanced by the primary radiation torque turning the disk parallel to the substrate and the secondary radiation torque aligning the disk perpendicular to the substrate. The particle velocity from secondary radiation is shown along the interparticle direction as: $\boldsymbol{v_{sec}} = -\frac{f_{0,l/w}}{3}\frac{p_0}{\rho_l c_l} k_l a\hat{\mathbf{r}}$. As the two-particle velocities are of similar magnitude, the equilibrium disk orientation is ~45° (135°) with the z-axis, consistent with the experimental results (Fig. 2d, Supplementary Movie 3) and dynamic simulation results (Fig. 4d, Supplementary Movie 6). With more droplets joining to form a bigger cluster, the particle velocity from secondary radiation was enhanced with all pairwise scattering from neighboring droplets (see Supplementary Section 4 for quantitative scaling of secondary radiation particle velocity for a trimer, Supplementary Fig. 9). Thus, the equilibrium disk orientation favored perpendicular alignment due to the dominating secondary effects, which is consistent with experimental observations (Fig. 2e), as well as numerical dynamic simulation (Fig. 4e, Supplementary Movie 4). This can be seen in

clusters of 3 or more droplets, where all of them showed similar perpendicular orientation of the disks (Fig. 2e–g).

**External control over the disk orientation**. From the equations for the primary and secondary radiation forces (Eq. 3 and Eq. 4), we realized that the primary radiation force is proportional to the wavenumber (proportional to frequency), whereas the secondary radiation force is proportional to the wavenumber raised to the power of 2. This implies that the secondary radiation force is more sensitive to the frequency of the wave than the primary radiation force. Also, from our discussion in the previous section, we can summarize that the primary radiation force is acting to push and rotate the disks to the top in a parallel position (parallel to substrate), whereas the secondary radiation force is acting to flip the disk to the perpendicular position (perpendicular substrate). Using the same logic, we tried manipulating the frequency of our acoustic wave in order to externally manipulate the orientation of the disks inside the droplets. For this purpose, a chirped IDT device (IDT device that can generate SAW in different frequencies) was fabricated that could generate SAW at both 10 and 20 MHz frequencies[23].

At a frequency of 10 MHz, due to the smaller secondary radiation force compared to the primary radiation force, the disks in all the clusters oriented parallel to the substrate (Fig. 5a, Supplementary Movie 7) compared to the disks being perpendicular at 20 MHz (Supplementary Movie 1). To analyze this effect of frequency, numerical simulations were performed on dimers and trimers at a frequency of 10 MHz. For dimers, the simulation results show that the equilibrium orientation angle for the disks was ~23° with the z-axis, which is much lower than the orientation angle at a frequency of 20 MHz (45°) (Fig. 5b vs. 4d, Supplementary Movie 3, Supplementary Movie 6). The effect of the difference in frequency on orientation behavior was clearly seen in 3-droplet clusters (Fig. 5a (ii), Supplementary Movie 8). To demonstrate the utility of this behavior, we repeatedly switched the frequencies from 20 to 10 MHz and vice versa and found tunable orientation of the disks from perpendicular (at 20 MHz) to parallel (at 10 MHz) and vice versa (Fig. 5c, Supplementary Fig. 10, Supplementary Movie 9). For better visualization of this phenomenon, we incorporated the cross-polarized microscopy (CPM)[44] technique to image the disks. Under CPM, due to the difference in the crystal orientation of the solid, the disk was only visible when the disks were oriented in the long axis (perpendicular to the substrate). Hence, at the frequency of 10 MHz (when disks were parallel), the field is dark, whereas when the frequency was changed to 20 MHz, the disks brighten up (Fig. 5c, Supplementary Movies 10 and 11). This behavior was reproducible and was observed for droplets within larger cluster sizes as well (Supplementary Fig. 10, Supplementary Movie 11). When intermediary frequencies were used (between 10 and 20 MHz), the frequency that switched the disk orientation from parallel to perpendicular varied with the number of droplets in the cluster as well (details in supplementary section 5, Supplementary Fig. 12). As our study focuses on the reorientation of the disks (seen between 10 and 20 MHz) and its mechanisms, we leave the detailed study of the intermediary frequencies for future investigations. This on-demand reorientation of the disks could in principle be used for applications involving acoustic-optical filters, as the total light intensity can be accurately switched from high to low as shown in Fig. 5d. Moreover, this blinking effect observed while repeatedly changing the frequencies could also be of interest for super-resolution imaging where deep underlying microvasculature can be mapped[45].

Additionally, in the field of microbiology, the ability to control internal structures can be of immense value. But currently,

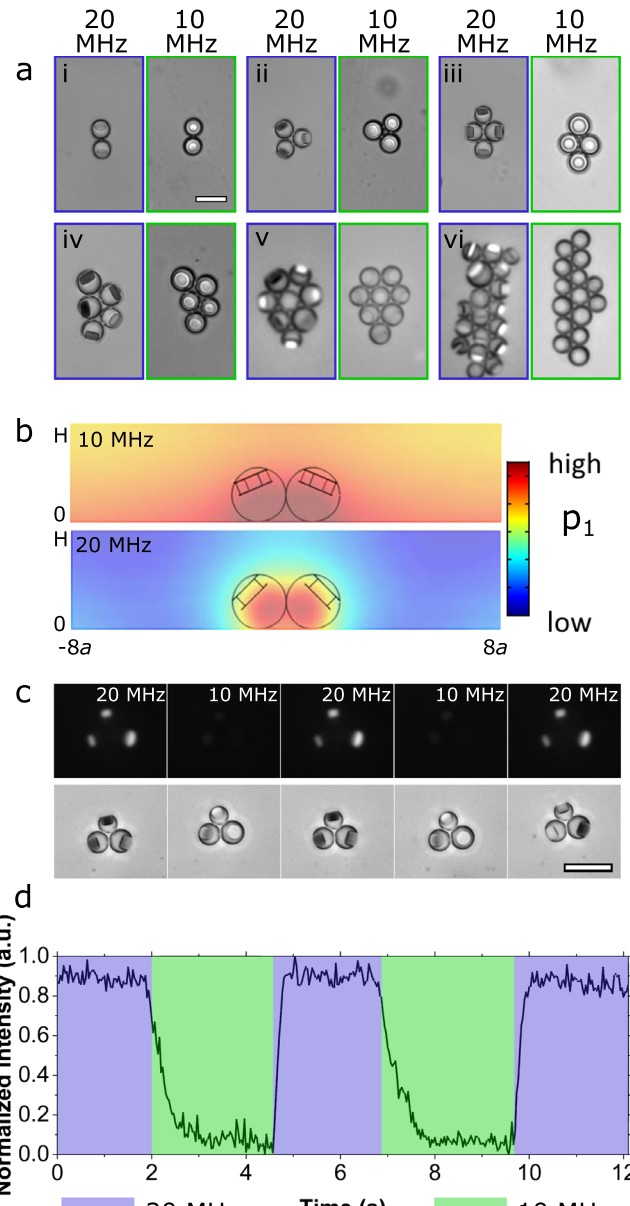

invasive techniques such as atomic force microscopy[46], optical traps[47] and magnetic tweezers[48] are used for manipulation and study the dynamics of the internal organelles of cells[49]. The technique demonstrated here in this study that has the ability to both move the particles themselves as well as the ability to manipulate the internal structures, can provide a noninvasive method to study the internal dynamics of the organelles within cells in a more natural environment. Similarly, recent studies of organoids, which are stem cell-derived 3D multicellular in-vitro tissue culture systems, have shown tremendous promise. Human organoids have been used to study various disease and cancer cells through genetic engineering of stem cells[50,51]. But, a major concern is that the ability of the organoid to self-organize is not enough to generate fully functional organs[52]. Moreover, the development of proper tissue in-vivo is also subject to external physical stimuli supplied at precise spatial order[52]. A technique like ours could be useful in this aspect as it can provide spatial control of the cluster as well as provide external physical forces, using acoustic radiation force, to mimic in-vivo conditions and aid to reproducibly generate organoids with a high level of

**Fig. 5 Effect of acoustic frequencies on the disk orientations. a** clusters of endoskeletal droplets with 2 (i), 3 (ii), 4 (iii), 5 (iv), 7 (v) and multiple (vi) droplets in the cluster at 20 MHz standing acoustic wave (left) and 10 MHz (right). Note that the solid disks are oriented parallel to the surface for 10 MHz whereas they are oriented perpendicular to the surface at 20 MHz. Scale bar, 20 μm. **b** Simulation results of dimer clusters in 10 MHz wave (top) and 20 MHz wave (bottom) showing total pressure distribution in the yz plane. The colors (red/blue) represent pressure amplitudes (high/low). Note that the pressure distribution between the two droplets (due to secondary radiation force) is more dramatic for 20 MHz than 10 MHz, which results in a larger tilting angle for the solid disks at 20 MHz. **c** Cross-polarized microscopy (top) images of a 3-droplet cluster where the disks are bright when they are perpendicular to the surface (20 MHz). Brightfield image of the same droplet cluster (bottom) shows the disk flipping between parallel and perpendicular just by changing input frequencies in a chirped IDT device (see Supplementary Movie 9 and 10). Larger droplet clusters are shown in Supplementary Fig. 10 and Supplementary Movie 11. Scale bar, 20 μm. **d** Plot corresponding to the normalized intensity (arbitrary units) from the cross-polarized microscopy images recorded (**c**) while continuously changing frequencies every ~2.5 s. Purple area denotes 20 MHz whereas green area denotes 10 MHz. Source data provided as a source data file.

maturation. Although this acoustic approach is superior to the optical method in terms of biocompatibility, toxicity and throughput, its demonstrated capability of manipulation, such as moving nanoparticles from one specific position to another inside a cell or droplet, is limited in comparision to optical ones and needs to be further improved if necessary.

From a materials perspective, it is well known that existing approaches in particle assembly only consider the interactions between the particles, the liquid medium and the external fields[53]. However, this technique renders the idea of a unique assembly system that incorporates the interactions between the external fields, the particle and its internal structure having opposite specific material properties from the particle body, and thus further expanding the design space for the assembly systems. In the next generation of assembly system, the assembly of particles with internal structures of different acoustic, magnetic, electric, or optical properties under external fields can be of great interest and studied for advanced assembly performance with potential applications in complicated dynamics of soft robotics and adaptive functional materials. In addition, such dynamic rotation behavior as function of input frequency provides a useful model system to study the advanced control strategies of assembly systems[54–56]. Current model systems exclusively focus on the control over the 2D positions of the particles, yet not spatial angles. The limitations would mainly come from the complexity of the synthesis of such internal structures, such as, finding compatible materials with desirable melting points, or finding compatible surfactants to generate desirable architectures of droplets. But with the help of the expanding knowledge of interfacial processes, the development of various types of surfactants with tunable interfacial energies[57], as well as the formation of various type of endoskeletal droplets[11], synthesizing different types of internal-structured droplets or particles is promising.

In summary, we demonstrated a unique acoustic manipulation of the internal structure of disk-in-sphere endoskeleton droplets, a very interesting yet challenging manipulation that will aid in our understanding of the effect of radiation forces. Under a one-dimensional standing acoustic wave, endoskeletal droplets move to the antinode and attract each other to form clusters. However, a repulsive secondary radiation force between the disks and drops

caused the disks to align perpendicularly to the substrate and perpendicularly to the droplet cluster centroid. This orientation of the internal disks could be reversibly manipulated by simply using different frequencies of acoustic wave, which changes the balance between the primary and secondary radiation forces responsible for the disk orientation. This reversible on-demand manipulation of the disk orientation can potentially be utilized in various filtering as well as imaging applications. This distinctive dynamic manipulation could potentially provide further opportunities for directed colloidal assembly with dynamic and acoustically tunable internal structures and pave the way towards manipulation of the internal structures of organoids and cells.

## Methods

**Materials.** The following chemicals were used as received: perfluorohexane (PFH) ($C_6F_{14}$, 99%, FluoroMed, Round Rock, TX, USA); perfluorododecane (PFDD) ($C_{12}F_{26}$, >99%, Fluoryx Labs, Carson City, NV, USA); 1,2-dibehenoyl-sn-glycero-3-phosphocholine (DBPC) (99%, Avanti Polar Lipids, Alabaster, AL, USA); N-(methylpolyoxyethyleneoxycarbonyl)-1,2-distearoyl-sn-glycero-3-phosphoethanolamine (DSPE-PEG5K) (NOF America, White Plains, NY, USA); poly-dimethylsiloxane (PDMS) (Sylgard 184 Silicone Elastomer, Dow, Midland, MI; positive photoresist (MEGAPOSIT SPR220-3.0, Dupont, Wilmington, DE); Tri-chloro(1H,1H,2H,2H-perfluorooctyl)silane, Polyvinyl Alcohol (PVA) (Sigma Aldrich, St. Louis, MO); photoresist developer ((MEGAPOSIT MF-26A, Dupont, Wilmington, DE; ultrapure deionized (DI) water from Millipore Direct-Q (Millipore Sigma, St. Louis, MO, USA).

**Preparation of the surfactant (lipid) solution.** The lipid solution was formulated by suspending DBPC and DSPE-PEG5K (9:1 molar ratio) at a total lipid concentration of 10 mg/mL in DI water. The lipids were first dissolved and mixed in chloroform in a glass vial, and then the solvent was removed to yield a dry lipid film at 35 °C and under vacuum overnight. The dry lipid film was rehydrated using DI water and then sonicated at 75 °C at low power (3/10) for 10 min to convert the multilamellar vesicles to unilamellar liposomes.

**Fabrication of flow focusing PDMS device.** Standard soft lithography techniques were used to construct the polydimethylsiloxane (PDMS) devices. Two device designs were prepared for the PDMS devices. One for the droplet generation and another one for SAW experiments. For both, masks for lithography were drawn using CleWin4 layout editor software (WeiWeb, Hengelo, The Netherlands) (Fig. 1a) and transparency masks were printed commercially (CAD/Art Services, Bandon, OR) at high resolution. To create the silicon mold, a layer of positive photoresist (MEGAPOSIT SPR220-3.0, Dupont, Wilmington, DE) was spin-coated on a silicon wafer (El-Cat Inc., Ridgefield Park, NK), pattern-transferred with a mask exposer (MJB4, KARL SUSS, Germany), and developed in a photoresist developer (MEGAPOSIT MF-26A, Dupont, Wilmington, DE). Afterwards, the substrate was dry-etched with SF6 plasma (PlasmaSTAR, AXIC Inc., Santa Clara, CA). The silicon mold was silanized by vapor deposition of Trichloro(1H,1H,2H,2H-perfluorooctyl)silane into the mold before use. PDMS pre-polymers (Sylgard 184 Silicone Elastomer, Dow, Midland, MI) were mixed (10:1 weight ratio of base:curing agent), degassed in a vacuum desiccator for 30 mins, cast into the silanized silicon mold and cured at 65 °C overnight. After curing, individual PDMS devices were cut to shape from the mold.

Final device (for droplet generation) was prepared by treating the glass slide and the precut PDMS device with air plasma using plasma wand (BD-10AS High-Frequency Generator, Electro Technic Products, Chicago, IL) for 30 s. The two surfaces were brought into contact for proper bonding. 1% PVA solution was put and left in the channels for 15 mins to make them hydrophilic[58]. The excess PVA solution was flushed out and the device was heated at 115 °C for 15 mins to vaporize any excess water in the channels. Devices were then heated at 65 °C overnight.

**Fabrication of interdigital transducers (IDTs).** 128° Y–X cut Lithium Niobate, LiNbO3, wafers were purchased (Precision Micro-Optics, Burlington, MA) and cleaned by sonicating in acetone, isopropyl alcohol and deionized (DI) water for 5 min each. The interdigitated electrodes (IDTs) were patterned by standard microfabrication techniques. Typically, the LiNbO3 was spin-coated with positive photoresist (S1813, thickness of ~1.5 μm), and exposed to UV light under a mask. The exposed photoresist was dissolved in MF-26A Developer. The IDTs were finally formed by E-beam evaporation (10 nm Cr, 100 nm Au) and lift-off processes. Furthermore, 300 nm of SiO2 was deposited on the substrate by magnetron sputtering to prevent corrosion of the IDTs and to enhance channel bonding. The IDTs consist of 20 finger-pairs with a 10 mm aperture and a 200 μm periodicity (50 μm finger width), and the resonance frequency was then measured using a Keysight E5061B vector network analyzer (VNA) at 19.8 MHz.

**Synthesis of endoskeletal droplets**. The solid (perfluorododecane, $C_{12}F_{26}$) and the liquid (perfluorohexane, $C_6F_{14}$) were mixed at a ratio of 45 mole percent solid. The mixture was heated until the solids melted (~45 °C) in a heated water bath while intermittently stirring the mixture in a vial mixer (Mini Vortexer, Fisher Scientific). The liquid mixture was put in a glass syringe (Gastight 1750, Hamilton, Reno, NV) that was continuously heated using a syringe heater (Syringe Heater, New Era Pump Systems, Farmingdale, NY) set at 50 °C to keep the FC mixture in liquid phase. This syringe was set in a syringe pump (Fusion 4000, Chemyx, Stafford, TX) and was connected to the FC inlet of the PDMS device using flexible plastic tubing (OD 0.07 inches, ID 0.04 inches, Tygon) and steel tube (18 G, 204 SS, Component Supply, Sparta, TN). The tubes were heated using a heat lamp (BR40 Incandescent Heat lamp, 125 W, GE) and focusing the heat on the tubes using curved aluminum foil. The PDMS device itself was placed on top of a flexible heater (Kapton KHLV-102/10-P, Omega Engineering, Norwalk, CT, USA) attached to a power supply (Agilent E3640A, Agilent Technologies, Santa Clara, CA, USA) which continuously heated the PDMS device. The PDMS device setup was mounted on a microscope (Olympus IX71, Olympus Life Sciences) and images/videos were recorded using a digital CCD camera (Qimaging QIClick digital CCD camera). The syringe containing lipid solution was set in another syringe pump (GenieTouch, Kent Scientific) and was connected to the lipid inlet of the PDMS device using the same plastic tubing and steel tube. The lipid solution and FC mixture were injected at flowrates of 20 µl/min and 1 µl/min respectively. The generated droplets were collected from the collection chamber (outlet) into a 2 ml glass vial and the emulsion was cooled to solidify the endoskeleton and stored in the fridge until used.

**Synthesis of liquid PFH droplets**. Liquid perfluorohexane (PFH, $C_6F_{14}$) droplets were generated using the same PDMS microfluidic device (Fig. 1a–c). PFH was set in a syringe pump and connected to the FC inlet. Lipid solution (similar to the earlier section) was used as the aqueous phase. The lipid solution and FC phase were injected at flowrates of 20 µl/min and 1 µl/min respectively.

**Surface acoustic wave directed assembly**. The PDMS channel was plasma treated before attaching it to the LiNbO3 device to treat the PDMS surface and make it hydrophilic. Endoskeletal droplet solution was diluted (10X) and 10 µl of the diluted solution was pipetted at the edge of the PDMS device channel such that the emulsion would flow into the device by capillary effect. When the channel was fully wetted, the two ends were sealed using vacuum grease (Dow Corning, Houston, TX, USA). The IDTs were connected to a RF Signal generator (SDG 5082, Silgent Technologies) and amplified by a power amplifier (403LA Broadband Power Amplifier, Electronics, and Innovation). The IDT device with the PDMS channel was mounted on an inverted microscope (Nikon Eclipse Ti2 Inverted Microscope, Melville, NY, USA) fitted with Nikon Plan Flour 4X, 10X and 40X objectives. The microscope was attached to a digital CMOS camera (Hamamatsu C11450 ORCA Flash-4.0LT, Bridgewater, NJ, USA). Images were acquired using a custom-built LabVIEW VI.

**Cross polarized microscopy**. Images and videos were recorded by using two polarizer filters at 90° with each other. One of the filters was placed before the sample and the other was placed after the sample in the light path of the inverted microscope.

## Data availability

Source data are provided with this paper. All data generated or analyzed during this study are included in this published article (and its supplementary information files and the source data file). Any additional data are available from the corresponding author on reasonable request. Source data are provided with this paper.

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

## Acknowledgements

The authors acknowledge funding support from the CU Boulder startup fund and W. M. Keck Foundation grant to X.D., and NIH grants R01CA195051 and R01HL151151 to M.B. G.S. acknowledges support from Thomas and Brenda Geers Fellowship. A.K.F acknowledges support from the Teets Family Endowed Doctoral Fellowship. The Microfluidic devices were fabricated in JILA clean room at University of Colorado Boulder. Publication of this chapter was partially funded by the University of Colorado Boulder Libraries Open Access Fund.

## Author contributions

G.S., T.Y, M.A.B and X.D. designed the experiments; X.D. conceived the research; M.A.B. and X.D. supervised the research; G.S. performed the experiments; T.Y. performed the simulations; Y.G. and A.K.F. fabricated the devices. G.S. and T.Y. analyzed data; G.S., T.Y, Y.G., A.K.F., B.L., M.R., M.A.B and X.D. discussed the research and wrote the paper.

## Competing interests

The authors declare no competing interests.
