## [Peer Review File · Nature Communications]

REVIEWER COMMENTS

Reviewer #1 (Remarks to the Author):

Acoustically Manipulating Internal Structure of Disk-in-Sphere Endoskeletal Droplets

Gazendra Shakya et al.

Review comments:

The authors explore the possibilities of manipulating the internal structure of the droplet using the acoustic wave for a disk-in-sphere endoskeletal droplet.

The authors have conducted a detailed experimental study to demonstrate the same and presented an elaborate theoretical analysis of the mechanisms behind the observations. The orientation of the disk was able to modulate with the acoustic driving frequency. The present work is of great interest and has potential utilities in various lab-on a chip devices. The manuscript is recommended for publication after addressing the minor comments mentioned below.

- The drag force F_{μ} acting on the droplet (Hadamard-Rybczynski formula) may be an important parameter influencing the entire physics of the problem. There was no discussion about such drag interaction in these system. Will such forces be significant in the present context? Also the authors may look into the relevance of Capillary number based on the interfacial energy for the migration dynamics.

- Also will the DLVO interaction be significant at such micro scale interaction?

- The details pertaining to the simulation may be included keeping in view of the reproducibility of the work such as the mesh refinement and details of grid independence etc. Also please comment on whether the attenuation in PDMS be significant or insignificant.

- The acoustic energy dissipation would essentially increase the internal energy of the system and subsequently the temperature. Authors may discuss how this issue has been considered in the present study. There some recent works on acoustothermal heating in such devices.

- It would be better if the authors can highlight and enumerate how the present findings would be of potential utility to various potential applications and advances over the present state of art to have better insight to the readers.

- Minor comment: The title of supplementary and main manuscripts are different.

Reviewer #2 (Remarks to the Author):

This paper introduced a method of manipulating the orientation of disks inside droplets using primary and secondary acoustic waves, and further probed into the fundamental physical reasons for the observed behavior. This allowed insight leading to application of methods like variable frequencies to “switch” skeletal disk orientation.

The work is interesting, thorough, coherent, and concise. While previous literature includes manipulation of shaped particles and various materials via acoustics, and manipulation of internal features via a variety of external applied fields, the manipulation of internal features via acoustic waves is quite new to my knowledge.

I observed no mathematical or conceptual errors in the text. In general the writing and figures are also well-prepared.

I had some minor suggestions to improve readability.

- 1) The arrow notation on Figure 1 is basically unreadable to me. I'm still not sure what it's trying to show. Having a much more zoomed-in image of what you're indicating could help, in addition to the large image

- 2) The paragraph after equation 4 could be clearer

- 3) In equation 4, add parentheses around delta or move f factors in front of cos so the notation is clear as to what argument the cosine function acts on

- 4) Text in Figure 3 is too small

- 5) The blue arrows in SI Fig 2 are very hard to discern - you might consider different annotation

- 6) General curious questions: In our lab we have observed “beating” effects in two-dimensional standing waves. Would you be able to use a method like this to cause disks to oscillate in interesting or useful ways? Do you have ways to reliably tune the number of droplets at a given node/antinode? Is there a way to “freeze” the disks in location and link a cluster together after they have been assembled and oriented?

Response to Reviewer Comments

Article: NCOMMS-21-30512

Title: "Acoustically Manipulating Internal Structure of Disk-in-Sphere Endoskeletal Droplets"

We thank the reviewers and editor for their constructive as well as positive comments. We added additional experiments and numerical results and addressed all the review and editorial comments which obviously helped improve our manuscript. Our point-by-point responses are given below:

Reviewer #1.

The authors explore the possibilities of manipulating the internal structure of the droplet using the acoustic wave for a disk-in-sphere endoskeletal droplet.

The authors have conducted a detailed experimental study to demonstrate the same and presented an elaborate theoretical analysis of the mechanisms behind the observations. The orientation of the disk was able to modulate with the acoustic driving frequency. The present work is of great interest and has potential utilities in various lab-on a chip devices. The manuscript is recommended for publication after addressing the minor comments mentioned below.

Response: We appreciate the positive comments.

1) The drag force F_{μ} acting on the droplet (Hadamard-Rybczynski formula) may be an important parameter influencing the entire physics of the problem. There was no discussion about such drag interaction in these system. Will such forces be significant in the present context? Also the authors may look into the relevance of Capillary number based on the interfacial energy for the migration dynamics.

Response: Great point. Past work has shown that drag force is dominant for smaller particles ($<2 \mu\text{m}$) whereas radiation force is dominant for larger particles (Qiu et. al¹, Muller et. al², Barnkob et. al³). Since our particles are much larger ($\sim 10 \mu\text{m}$) and the particle velocity is dominated by the acoustic radiation force, the drag force would not be significant in the present context. The manuscript was modified to include the following explanation (line 43-47):

"In an acoustic field, motion of the particles (much smaller than the wavelength of the acoustic wave) suspended in an aqueous medium is driven by two main acoustofluidic forces: acoustic radiation force, which is responsible for the motion of larger particles ($> 2 \mu\text{m}$), and the stokes drag force arising from acoustic streaming, which is responsible for the motion of much smaller particles ($< 2 \mu\text{m}$)."

2) Also will the DLVO interaction be significant at such micro scale interaction?

Response: In our particular case, the DLVO interaction would not be significant as well because the aggregation behavior of droplets and the change in orientation observed in the disks is only observed when the SAW device is turned on. Without SAW we don't see any aggregation or repulsion or any specific orientation of the solid disk even when the disks come close to each other stochastically. So, we believe it is safe to assume that the behavior observed is due to the acoustic field generated by the standing SAW.

3) The details pertaining to the simulation may be included keeping in view of the reproducibility of the work such as the mesh refinement and details of grid independence etc. Also please comment on whether the attenuation in PDMS be significant or insignificant.

Response: We totally agree with the reviewer that the mesh refinement section was missing in the previous manuscript and now we have added the detailed meshing description in Supplementary Section 2 including a mesh refinement test (Supplementary Fig. 11) to justify our simulation result accuracy. As for the wave attenuation degree inside PDMS, it can be calculated via $\text{attenuation} = \alpha fL$, where α is the attenuation coefficient, f is the wave frequency and L is the PDMS thickness (distance along wave propagation), correspondingly. The wave attenuation through a 100 mm PDMS channel can be calculated as 0.56 dB, thus negligible as αf was measure as 56.09 dB/cm @ 10MHz by Tsou, et al⁴.

We clarified mesh settings on page 6 in Supplementary Materials. This will help the duplication of our results. Mesh refinement test in Supplementary Fig. 11 proved our simulation results accuracy. The manuscript was modified to include the following text (page 6 in SI):

“The computation mesh is composed of a Free Tetrahedral meshing composing liquid droplets plus inner disks and Boundary layers mesh with 5 boundary layers of a total thickness 0.5 mm composing rectangular channel surfaces. The rest of meshing is confined by a maximum element size of $\lambda_w/5$ as recommended by the COMSOL manual. The maximum element size of Free tetrahedral was chosen as $a/6$ as justified by the mesh refinement test in Supplementary Fig. 11.”

4) The acoustic energy dissipation would essentially increase the internal energy of the system and subsequently the temperature. Authors may discuss how this issue has been considered in the present study. There some recent works on acoustothermal heating in such devices.

Response: We agree with the reviewer, the acoustic energy dissipation would essentially increase the temperature of our channel, but this increase is only significant at longer periods of times and/or at a larger amplitude. Our experiments last anywhere between a few seconds to 10s of seconds over which the increase in temperature is minimal at such low power level⁵. Additionally, significant increases in the temperature of endoskeletal droplets can be visualized optically where the solid disks start melting and gets smaller until it fully melts and forms a uniform liquid droplet⁶. We did not observe this internal disk melting during any of our experimental runs, hence it is safe to say the temperature increase is not significant in the present study.

5) It would be better if the authors can highlight and enumerate how the present findings would be of potential utility to various potential applications and advances over the present state of art to have better insight to the readers.

Response: This is a great suggestion. The main utility of this work is that we have tried to enhance the understanding of the radiation forces and their effect on particle manipulation. This distinctive dynamic manipulation could potentially provide further opportunities for directed colloidal assembly with dynamic and acoustically tunable internal structures and pave the way towards manipulation of the internal structures of organoids and intracellular organelles. Particularly, we have included two additional paragraphs to further discuss and highlight the potential utility of the current findings in the manuscript. The manuscript was modified to include the following text in page 11 line 238, page 11 line 241-273, page 12 line 275 and page 12 line 281:

“Moreover, this blinking effect observed while repeatedly changing the frequencies could also be of interest for super resolution imaging where deep underlying microvasculature can be mapped.”

“Additionally, in the field of microbiology, the ability to control internal structures can be of immense value. But currently, invasive techniques such as atomic force microscopy, optical traps and magnetic tweezers are currently used for manipulation and study the dynamics of the internal organelles of cells. The technique demonstrated here in this study, that has the ability to both move the particles themselves as well as the ability to manipulate the internal structures, can provide a non-invasive method to study the internal dynamics of the organelles within cells in a more natural environment. Similarly, recent studies of organoids, which are stem cell derived 3D multicellular in-vitro tissue culture systems, have shown tremendous promise. Human organoids have been used to study various disease and cancer cells through genetic engineering of stem cells. But, a major concern is that the ability of the organoid to self-organize is not enough to generate fully functional organs. Moreover, development of proper tissue in-vivo is also subject to external physical stimuli supplied at precise spatial order. A technique like ours could be useful in this aspect as it can provide spatial control of the cluster as well as provide external physical forces, using acoustic radiation force, to mimic in-vivo conditions and aid to reproducibly generate organoids with a high level of maturation. Although this acoustic approach is superior to optical method in terms of biocompatibility, toxicity and throughput, its demonstrated capability of manipulation, such as moving nano particles from one specific position to another inside a cell or droplet, is limited in comparing to optical ones and needs to be further improved if necessary.

From a materials perspective, it is well known that existing approaches in particle assembly only consider the interactions between the particles, the liquid medium and the external fields. However, this technique renders the idea of a unique assembly system that incorporates the interactions between the external fields, the particle and its internal structure having opposite specific material properties from the particle body, and thus further expanding the design space for the assembly systems. In the next generation of assembly system, the assembly of particles with internal structures of different acoustic, magnetic, electric, or optical properties under external fields can be of great interest and studied for advanced assembly performance with potential applications in complicated dynamics of soft robotics and adaptive functional materials. In addition, such dynamic rotation behavior as function of input frequency provides a new model system to study the advanced control strategies of assembly systems. Current model systems exclusively focus on the control over the 2D positions of the particles, yet not spatial angles. The limitations would mainly come from the complexity of the synthesis of such internal structures, such as, finding compatible materials with desirable melting points, or finding compatible surfactants to generate desirable architectures of droplets. But with the help of the expanding knowledge of interfacial processes, development of various types of surfactants with tunable interfacial energies⁵⁷ as well as the formation of various type of endoskeletal droplets, synthesizing different types of internal-structured droplets or particles is promising.”

“... manipulation that will aid in our understanding of the effect of radiation forces.”

“This reversible on-demand manipulation of the disk orientation can potentially be utilized in various filtering as well as imaging applications.”

6) Minor comment: The title of supplementary and main manuscripts are different.

Response: The title of the supplementary has been changed to match the main manuscript.

Reviewer #2

This paper introduced a method of manipulating the orientation of disks inside droplets using primary and secondary acoustic waves, and further probed into the fundamental physical reasons for the observed behavior. This allowed insight leading to application of methods like variable frequencies to “switch” skeletal disk orientation.

The work is interesting, thorough, coherent, and concise. While previous literature includes manipulation of shaped particles and various materials via acoustics, and manipulation of internal features via a variety of external applied fields, the manipulation of internal features via acoustic waves is quite new to my knowledge.

I observed no mathematical or conceptual errors in the text. In general the writing and figures are also well-prepared.

I had some minor suggestions to improve readability.

Response: We appreciate the positive comments and address all the suggestions accordingly.

1) The arrow notation on Figure 1 is basically unreadable to me. I'm still not sure what it's trying to show. Having a much more zoomed-in image of what you're indicating could help, in addition to the large image

Response: Zoomed in images of the droplets with parallel orientation (disk seen as circles) and perpendicular orientation (disk seen as rectangles) of the disks have been added to figure 1 as suggested by the reviewer. We hope this clarifies our definition of parallel and perpendicular orientation of the disks.

2) The paragraph after equation 4 could be clearer

Response: We have added some clarifying phrases to the paragraph proceeding equation 4. We hope those changes, which are highlighted in the manuscript line 134-139, will make the paragraph clearer to our readers.

3) In equation 4, add parentheses around delta or move f factors in front of cos so the notation is clear as to what argument the cosine function acts on

Response: Equation 4 has been updated as suggested by the reviewer.

4) Text in Figure 3 is too small

Response: The texts in figure 3 has been enlarged as suggested by the reviewer. We hope it aids in the readability of the figure.

5) The blue arrows in SI Fig 2 are very hard to discern - you might consider different annotation

Response: The arrows in SI Fig 2 have been enlarged per suggestion and we hope it aids in the readability of the figure.

6) General curious questions: In our lab we have observed “beating” effects in two-dimensional standing waves. Would you be able to use a method like this to cause disks to oscillate in interesting or useful ways? Do you have ways to reliably tune the number of droplets at a given node/antinode? Is there a way to “freeze” the disks in location and link a cluster together after they have been assembled and oriented?

Response: That could be a very interesting discussion. We have not yet looked at the behavior of these droplets under 2D standing wave as the current study just focuses on 1D standing waves. We have not observed such “beating” effects in 1D standing waves, but we would be more than happy to explore the disk behavior in 2D standing wave in our future work. Would the “beating” be possibly because of the disequilibrium of acoustic streaming (induced hydrodynamic force) and radiation force? Sometimes acoustic force may push particle towards one location while the acoustic streaming push it towards another location. Currently, the number of droplets in a cluster cannot be ideally tuned. It just depends on the concentration of droplets in the aqueous chamber and the proximity of one droplet to another in the node/antinode region. We can only statistically tune the number of droplets at given nodes/antinodes by simply adjusting the total concentration of the droplets. There is a way to link the cluster of droplets together after the droplets have been assembled and oriented. For example, people have demonstrated the feasibility of linking droplets via functionalizing the emulsion surface with DNA molecules⁸. Since the disk is fixed inside the droplet under acoustic field (as long as the parameters are kept the same), the location of the disk should thus be “frozen” after the cluster is linked. However, freezing disks in situ is more challenging as it involves solid/liquid interface. Currently, we could not find an effective way to do so, to the best of our knowledge.

Editor Comments

We would also ask you to consider expanding your demonstrations to a larger number of frequencies (i.e. 5, 15, 25 Hz) and its impact in the droplets configurations, which is needed to grasp the range of actuation control that can really be achieved.

Response: We appreciate this valuable suggestion, assuming the suggested frequencies are in the unit of MHz (5, 15, 25 MHz). Per Editor’s request, we conducted additional experiments of the disk dynamics under 10, 12, 14, 16, 18, and 20 MHz. We discuss these results in the main manuscript in page 10 line 232:

“When intermediary frequencies were used (between 10 and 20 MHz), the frequency that switched the disk orientation from parallel to perpendicular varied with the number of droplets in the cluster as well (details in supplementary section 5). As our study focuses on the reorientation of the disks (seen between 10 and 20 MHz) and its mechanisms, we leave the detailed study of the intermediary frequencies for future investigations.”

We also added a new section (section 5) in supplementary information and added a new figure (supplementary figure 12) to further discuss these results:

“Note that the current work focuses on the droplet aggregation and disk orientation of at a frequency of 10 and 20 MHz as these two frequencies show us the two extremes of the

disk orientation behavior. At 10 MHz, even very large clusters show parallel orientation (figure 5 A) whereas in 20 MHz, 2 droplets clusters show the threshold transition between the parallel and perpendicular orientation of the disks (Fig 2D). This perpendicular orientation is brought about by the effect of the secondary radiation force. The effect of the secondary radiation force can be increased in two ways. First by increasing the frequency (equation 4), and second by increasing the number of droplets in a cluster (equation S10). Hence, smaller cluster of droplets would require higher frequencies for the disks to be perpendicular whereas disks in larger cluster of droplets would require lower frequencies for the disks to be perpendicular. To look into the effects of the number of droplets we performed similar experiments at various intermediary frequencies (12, 14, 16 and 18 MHz) using a chirped IDT device. The results are shown in Supplementary Figure 12 where we observed that for a 3-droplet cluster, most disks are perpendicular at 18-20 MHz, for a 4-droplet cluster, most disks are perpendicular at 16-18 MHz and finally for a 5-droplet cluster, most disks are perpendicular at 14-16 MHz.

Since our goal with this paper was to show on demand flipping of the disks from parallel to perpendicular, 10 and 20 MHz works well as it shows the two extremes of the disk arrangement behaviors for all the droplet clusters. Moreover, it is obvious, based on the investigation of the principles behind such phenomenon in this work, that disk orientation transition occurs between 10-20 MHz, as shown in the additional experimental observation in Supplementary Figure S12.”

Supplementary Figure 1: Intermediary frequencies. Series of images with 3 droplets cluster at top, 4 droplet cluster in second row and 5 droplet cluster at the bottom shown at different intermediary frequencies. Note that larger droplet clusters require lower frequencies for the disks to be perpendicular.

References:

1. Qiu, W., Bruus, H. & Augustsson, P. Particle-size-dependent acoustophoretic motion and depletion of micro- and nano-particles at long timescales. *Physical Review E* **102**, (2020).
2. Muller, P. B. *et al.* Ultrasound-induced acoustophoretic motion of microparticles in three dimensions. *Phys. Rev. E* **88**, 023006 (2013).
3. Barnkob, R., Augustsson, P., Laurell, T. & Bruus, H. Acoustic radiation- and streaming-induced microparticle velocities determined by microparticle image velocimetry in an ultrasound symmetry plane. *Physical Review E* **86**, (2012).
4. Tsou, J. K., Liu, J., Barakat, A. I. & Insana, M. F. Role of Ultrasonic Shear Rate Estimation Errors in Assessing Inflammatory Response and Vascular Risk. *Ultrasound in Medicine & Biology* **34**, 963–972 (2008).
5. Ding, Y. *et al.* On-Chip Acousto Thermal Shift Assay for Rapid and Sensitive Assessment of Protein Thermodynamic Stability. *Small* **16**, 2003506 (2020).
6. Shakya, G. *et al.* Vaporizable endoskeletal droplets via tunable interfacial melting transitions. *Sci Adv* **6**, eaaz7188 (2020).
7. Liu, J., Wen, J., Zhang, Z., Liu, H. & Sun, Y. Voyage inside the cell: Microsystems and nanoengineering for intracellular measurement and manipulation. *Microsyst Nanoeng* **1**, 15020 (2015).
8. McMullen, A., Holmes-Cerfon, M., Sciortino, F., Grosberg, A. Y. & Brujic, J. Freely Jointed Polymers Made of Droplets. *Phys. Rev. Lett.* **121**, 138002 (2018).

REVIEWERS' COMMENTS

Reviewer #1 (Remarks to the Author):

The authors have addressed the comments satisfactorily. The manuscript may be accepted for publication in the present format.

Reviewer #2 (Remarks to the Author):

All of my comments have been addressed.